# Novel Biomarkers for Inflammatory Bowel Disease and Colorectal Cancer: An Interplay between Metabolic Dysregulation and Excessive Inflammation

**DOI:** 10.3390/ijms24065967

**Published:** 2023-03-22

**Authors:** Mohamed Salla, Jimmy Guo, Harshad Joshi, Marilyn Gordon, Hitesh Dooky, Justine Lai, Samantha Capicio, Heather Armstrong, Rosica Valcheva, Jason R. B. Dyck, Aducio Thiesen, Eytan Wine, Levinus A. Dieleman, Shairaz Baksh

**Affiliations:** 1Department of Biochemistry, Faculty of Medicine & Dentistry, University of Alberta, 113 Street 87 Avenue, Edmonton, AB T6G 2E1, Canada; 2Department of Medicine, Faculty of Medicine & Dentistry, University of Alberta, 113 Street 87 Avenue, Edmonton, AB T6G 2E1, Canada; 3Department of Pediatrics, Faculty of Medicine & Dentistry, University of Alberta, 113 Street 87 Avenue, Edmonton, AB T6G 2E1, Canada; 4Department of Oncology, Faculty of Medicine & Dentistry, University of Alberta, 113 Street 87 Avenue, Edmonton, AB T6G 2E1, Canada; 5Department of Pharmacology, Faculty of Medicine & Dentistry, University of Alberta, 113 Street 87 Avenue, Edmonton, AB T6G 2E1, Canada; 6Cardiovascular Research Centre, Faculty of Medicine & Dentistry, University of Alberta, 113 Street 87 Avenue, Edmonton, AB T6G 2E1, Canada; 7Laboratory Medicine and Pathology, Faculty of Medicine & Dentistry, University of Alberta, 113 Street 87 Avenue, Edmonton, AB T6G 2E1, Canada; 8Department of Physiology, Faculty of Medicine & Dentistry, University of Alberta, 113 Street 87 Avenue, Edmonton, AB T6G 2E1, Canada; 9Cancer Research Institute of Northern Alberta and Women and Children’s Health Research Institute, 113 Street 87 Avenue, University of Alberta, Edmonton, AB T6G 2E1, Canada; 10BioImmuno Designs, Inc., 4747 154 Avenue NW, Edmonton, AB T5Y 0C2, Canada

**Keywords:** IBD, CRC, AMPK, insulin, metformin, resveratrol, RASSF1A, YAP, RIPK2, dextran sulphate sodium

## Abstract

Persistent inflammation can trigger altered epigenetic, inflammatory, and bioenergetic states. Inflammatory bowel disease (IBD) is an idiopathic disease characterized by chronic inflammation of the gastrointestinal tract, with evidence of subsequent metabolic syndrome disorder. Studies have demonstrated that as many as 42% of patients with ulcerative colitis (UC) who are found to have high-grade dysplasia, either already had colorectal cancer (CRC) or develop it within a short time. The presence of low-grade dysplasia is also predictive of CRC. Many signaling pathways are shared among IBD and CRC, including cell survival, cell proliferation, angiogenesis, and inflammatory signaling pathways. Current IBD therapeutics target a small subset of molecular drivers of IBD, with many focused on the inflammatory aspect of the pathways. Thus, there is a great need to identify biomarkers of both IBD and CRC, that can be predictive of therapeutic efficacy, disease severity, and predisposition to CRC. In this study, we explored the changes in biomarkers specific for inflammatory, metabolic, and proliferative pathways, to help determine the relevance to both IBD and CRC. Our analysis demonstrated, for the first time in IBD, the loss of the tumor suppressor protein Ras associated family protein 1A (RASSF1A), via epigenetic changes, the hyperactivation of the obligate kinase of the NOD2 pathogen recognition receptor (receptor interacting protein kinase 2 [RIPK2]), the loss of activation of the metabolic kinase, AMP activated protein kinase (AMPKα1), and, lastly, the activation of the transcription factor and kinase Yes associated protein (YAP) kinase, that is involved in proliferation of cells. The expression and activation status of these four elements are mirrored in IBD, CRC, and IBD-CRC patients and, importantly, in matched blood and biopsy samples. The latter would suggest that biomarker analysis can be performed non-invasively, to understand IBD and CRC, without the need for invasive and costly endoscopic analysis. This study, for the first time, illustrates the need to understand IBD or CRC beyond an inflammatory perspective and the value of therapeutics directed to reset altered proliferative and metabolic states within the colon. The use of such therapeutics may truly drive patients into remission.

## 1. Introduction

Inflammation is a complex defense mechanism against biological and chemical insults. Although beneficial, persistent inflammation can cause cellular damage, resulting in many diseases, including IBD and CRC. Inflammation is an essential immune response that involves controlled activation of NFκB and production of cytokines, promoting healing of damaged epithelial cells and defense against pathogenic agents. However, chronic inflammation of the gastrointestinal (GI) tract occurs with associated symptoms of diarrhea, abdominal pain, and weight loss. As many as 1 in 150 Canadians are diagnosed with IBD, with the prevalence for IBD in the world at 1 case per 250 persons, annually [1,2,3]. Several IBD patients will develop colorectal cancer (CRC) in less than 20 years. IBD includes ulcerative colitis (UC) and Crohn’s disease (CD), but the root cause of IBD is unknown. Current research suggests a combination of genetic predisposition, enhanced autophagic response, epigenetic modulation, and microbiome disruptions may be involved [4]. Current IBD medications include aminosalicylic acids, disease-modifying antirheumatic drugs (DMARDs), steroids, biologic medications, and novel small molecule drugs [5,6].

Drivers of IBD include a complex interplay of environmental, dietary, genetic, and microbial determinants, that will initiate and perpetuate inflammation of the gastrointestinal tract [7]. It can occur at any age, from early childhood to adulthood, and can have life-long complications. It presents with diarrhea (often bloody), abdominal cramping, nausea, and overall discomfort. Severe cases will lead to hemi- or total colectomy of part or all of the colon, due to colonic erosion caused by enhanced inflammation. Reasons for surgical resection are many: some include fistulizing, stricturing disease, perforation, toxic mega-colon, and severe inflammation not responding to medical therapy [8]. Therapeutic intervention in treating IBD has greatly benefited from the understanding of the interplay of the innate and adaptive immune systems, identifying novel biomarkers of disease pathogenesis, the mapping of genetic factors predisposing individuals to IBD, and to the realization that our microbiome is a major determining factor of disease severity.

Numerous genes have been identified as initiating or driving the pathogenesis of IBD and thus influencing intestinal homeostasis [9,10]. These genes include, but are not limited to, cytokine factors (TNF-α, IL-10), cytokine receptors (IL-17R and IL-23R), transcription factors (NFκB and JAK/STAT pathway), kinases (PTPN22, Tyk2, RIPK2), apoptotic elements (CARD9 and caspase 11), and elements involved in autophagic signaling (ATG16L, IRGM, NOD2) [11]. Innate immunity provides a first-line of defense against not only invading microbial insults, but also tissue damage, and functions to activate tissue repair, inflammation, and microbial clearance. The surface of the GI tract is covered by a single layer of epithelial cells that functions as a physical barrier to the microflora found in the lumen, and is continually being stimulated by the natural microflora in the gut, and thus can develop tolerance to some insults. It can be disturbed/damaged during the pathogenesis of inflammatory disorders such as IBD [12,13,14]. Active IBD is characterized by pronounced infiltration of the lamina propria with innate immunity cells (macrophages, dendritic, and natural killer cells), as well as a later phase of infiltration of adaptive immune cells (B and T lymphocytes) stimulating the production of T regulatory cells (Treg), Th1, Th2, and Th17 cytokines [14,15,16]. Following inflammatory directed injury, the intestinal epithelial cell layer integrity needs to be re-established in a timely manner by the process of epithelial restitution (or resealing of the epithelial barrier), wound healing, and/or increased epithelial proliferation [17,18,19]. This will avoid a direct exposure of the lamina propria immune cells to the intestinal microflora and activation of an unnecessary inflammatory response [20].

There were more than two million new CRC cases diagnosed, and about one million CRC-related deaths in 2020, worldwide, representing 10% of the global cancer incidence and cancer related deaths. Although total cases of CRC have been declining worldwide, at a rate of 3% per year since 1990, it is surprising that there is an increase in CRC, by more than 2% per year since 1992, in individuals below 50 years of age. Furthermore, it is estimated that about 40% of CRC patients will die from their disease each year [21]. At diagnosis, more than 20% of CRC patients already have established metastases [4,22,23] and it is known that CRC can spread to common metastatic sites such as lymph nodes, liver, lungs, and peritoneum. The majority of patients are asymptomatic during early-stage CRC, when diagnosed as a result of screening. Thus, symptomatic presentation usually reflects relatively advanced CRC. 

Colonoscopy is the most accurate and versatile diagnostic test for CRC, and treatment options for CRC patients include radiation therapy, surgical removal of the tumor followed by adjuvant chemotherapy, and then targeted therapy using antibody-based therapies or small molecule inhibitors. Most CRC tumors arise within pre-existing adenomas, which harbor some of the genetic fingerprints of malignant lesions. Appearance of malignant lesions can take 10–15 years to arise (a clinical “remission period”), giving clinicians a window of opportunity to screen and subsequently remove these premalignant or early malignant lesions [24]. Thus, there is a need to better understand the inflammatory mediators/molecular drivers involved in promoting the pathogenesis of IBD-related CRC. 

In this study, we searched for biomarkers of IBD and CRC that also appeared in IBD-CRC case study patients. In addition, we explored the use of several animal models susceptible to inflammation injury, including a knockout of the tumor suppressor protein, Ras association domain family protein 1A (RASSF1A). The RASSF family of proteins contains ten related family members [25,26,27]. 1A is a tumor suppressor gene epigenetically silenced in cancer, without epigenetic loss of the other isoforms of RASSF1. RASSF6 and 8 may be involved in modulating NFκB by unknown mechanisms [28,29]. Direct association with K-Ras has been only observed for RASSF2, 4, 5A, 6, and 9 [30,31,32]. *Rassf1a^−/−^* mice are viable, fertile, and retain expression of isoform 1C and other RASSF gene family members. They have increased tumor incidence by 12–16 months of age (especially in the breast, lung, gastrointestinal tract, and immune system, e.g., B-cell related lymphomas) and develop tumors in response to chemical carcinogens [33,34]. Beyond six months, we have observed spontaneous colitis-like phenotype in *Rassf1a^−/−^* mice, that was accompanied by increased cytokine production (unpublished observations), indicating a possible role for 1A in regulating inflammation. Several reports indicate a role for 1A in mitosis, linked to its co-localization on microtubules, influencing the anaphase-promoting-complex [35,36,37,38], and links to centrosomes/spindle body during mitosis [38,39]. 1A is epigenetically silenced in IBD patients [40], which undermines its ability to restrict NFκB activation and prevent uncontrolled intestinal inflammation [41].

Our analysis revealed the importance of four biomarkers that were involved in tumor suppression, proliferation, inflammation, and metabolism within the colon, to suggest the need to find therapeutics to IBD that will reset abnormal inflammation, metabolism, proliferation, and epigenetic silencing, in order to drive patients into full remission. 

## 2. Results

### 2.1. The Tumor Suppressor Gene, RASSF1A, Is Epigenetically Silenced in IBD

For this study, we collected over 500 blood samples and >300 biopsy samples from pediatric, adult IBD, and non-IBD/control patients, by attending regular clinic appointments at the University of Alberta hospitals. Patient demographics and disease sub-types are outlined in Table 1 and Table 2 of the Section 4.2. We explored both epigenetic silencing of RASSF1A in patients with IBD, and if the intracellular NOD2 pathogen recognition receptor/RIPK2 pathway may be driving the inflammation in the colon of IBD patients. Previously, we utilized two pyrosequencing assays, covering 32 CpGs in the RASSF1A promoter, to explore the epigenetic silencing of RASSF1A in numerous cancers, including CRC [42]. That analysis revealed hotspots for epigenetic silencing between CpG1 and 8, that were also observed in a CRC patient that had liver metastasis [42], to suggest similar points of origin (the “CRC epigenetic signature”). Interestingly, analysis in peripheral blood of non-IBD, UC, and CD patients revealed a similar CRC epigenetic signature, with an epigenetic hotspot between CpG1 and CpG8 (Figure 1A, circled), to suggest similarity to CRC. In fact, we should rename it, “IBD-CRC epigenetic signature” now that we have isolated a similar hotspot in IBD. These analyses would then suggest a loss of expression of RASSF1A in tissue sections from IBD patients. 

We evaluated the expression of RASSF1A in intestinal biopsies from non-IBD, UC, and CD patients by immunohistochemical staining. Using an in-house developed monoclonal antibody to RASSF1A, we observed robust staining of descending colon sections from non-IBD patients, but reduced or no staining in UC or CD patients (Figure 1B), whereby the RASSF1A positive staining is reduced by >50% in most IBD patients (Figure 1B). This supports our epigenetic data to confirm the significant loss of RASSF1A, a tumor suppressor gene, in the descending colon of IBD patients. RASSF1A epigenetic inactivation can thus be observed in both cancers and inflammatory diseases such as IBD, and may be a robust molecular driver of IBD-related CRC.

### 2.2. The NOD2/RIPK2 Intracellular Pathogen Pathway Is a Molecular Driver of Inflammation in IBD

Inflammation is characterized by the hyperactivation of transcription factors (such as NFκB) through multiple pathways (both classical and non-classical), that includes TNF-R1 and the pathogen recognition pathway involving Toll like receptors (TLR) [43,44,45,46,47] and NOD2, an intracellular pattern recognition receptor. NOD2 is mainly stimulated by bacterial products containing muramyl dipeptide (MDP), and requires the obligate kinase RIPK2, to promote an autophagic response or a non-classical NFκB activation response [48]. Mice with genetic disruption of the *Nod/Ripk2*, have a dysbiotic intestinal flora, resulting in altered susceptibility to intestinal inflammation [49], as well as increased joint inflammation [50]. In addition, the loss of *Ripk2* has been demonstrated to result in the inability of cells to carry out mitophagy, leading to enhanced mitochondrial production of superoxide/reactive oxygen species, and accumulation of damaged mitochondria, that will trigger a caspase-11-dependent inflammasome activation [51,52]. We have previously published that, similar to most solid cancers, robust methylation of *RASSF1A* in inflammatory breast cancer (IBC) patients correlates with loss of expression [53]. Furthermore, we can observe a positive correlation between active RIPK2 (as monitored by RIPK2 pY474 antibody) and methylation status of *RASSF1A* in IBC tumor samples [53], to suggest expression loss of *RASSF1A* with increased levels of *RASSF1A* CpG methylation, and increased activation of active RIPK2. RIPK2 is controlled by complex posttranslational modification events, including autophosphorylation at several sites, including phosphorylation on tyrosine (Y) at position 474 and serine (S) 176 [54,55,56]. Phospho(p)-S176 RIPK2 antibodies do not perform well in detecting active RIPK2 on tissue sections, and thus we created our own RIPK2 antibody to detect the tyrosine (Y) phosphorylation at amino acid 474, that promotes an active RIPK2 (our RIPK2 pY474 antibody). This has been proven to work in both human and animal tissues. As obtained for IBC, IBD patients also had robust detection of active RIPK2 (Figure 2A), at >2 fold, in biopsy sections from IBD patients vs. non-IBD patients, suggesting its importance in driving colonic inflammation in IBD patients. Furthermore, elevated inflammation was confirmed in these tissue sections by analysis of myeloperoxidase (MPO) activity in the descending colon tissue samples from IBD patients (Figure 2B), to reveal similar fold changes in MPO activity in IBD when compared to non-IBD. 

### 2.3. Both AMPKα1 Activity and Insulin Production Are Altered Metabolic Parameters in IBD Patients

Metabolic pathways are influenced by numerous factors, ranging from stress, exercise, diet, genetics, and the gut microbiota [57]. Cancer arises due to unique reprograming of cells, to switch from aerobic respiration to rely more on glycolysis (known as the Warburg Effect) [58,59,60,61]. This is needed due to the hypoxic environment that most cancer cells find themselves in and the need to survive. Metabolic distress syndrome in IBD patients is understudied, but the AMP activated protein kinase (AMPK) [62,63,64,65,66] and mTOR pathway components have been suggested to be important [67]. In addition, the gut microbiota can control fatty-acid oxidation in the host, via suppression of the AMPKs. Interestingly, a common diabetic drug targeting AMPK, metformin, has been documented to reduce the incidence of CRC in diabetics [63,68]. Thus, we are beginning to realize that the link between metabolism, IBD-CRC, and the microbiome, may significantly contribute to disease progression towards malignancy. 

AMPK is a heterotrimeric fuel-sensing enzyme, that is activated by decreases in a cell’s energy state. When activated, it initiates metabolic and genetic events, that restore ATP levels by stimulating processes that generate ATP (e.g., fatty acid oxidation) and inhibiting others that consume ATP, but are not acutely required for survival (e.g., triglyceride and protein synthesis, cell proliferation [69]). When ATP levels fall, there is a corresponding increase in intracellular AMP levels, and AMPK is activated both allosterically by AMP and by phosphorylation of the catalytic subunit (α) on threonine (T) 172 by an upstream AMPK-kinase, LKB1. We carried out analysis of active AMPK by using the specific phospho-antibody, (AMPKα1) pT172. Surprisingly, we observed a significant > 50–70% loss of detection in IBD patients of active AMPK in the descending colon sections, shown in Figure 3A, to suggest an importance of AMPK to the metabolic stability of the colon. Equally as surprising, we detected insulin production in non-IBD colonic sections of IBD patients, that is almost completely lost in both CD and UC patients (Figure 3B). Insulin production in the colon has been reviewed in 2001 [70,71], suggesting that gut insulin may be involved in the response of the gastrointestinal tract to food. Furthermore, it was reported in 2019, that insulin may be able to promote cell death and transport in colon cancer [72]. Our analysis clearly revealed altered metabolism in the colon of IBD patients, related to the loss of active AMPK and insulin production, two outcomes that will significantly contribute to the metabolic distress syndrome in IBD patients and affect their ability to recover.

We subsequently explored the ability to regain a normal metabolic state, in terms of normalized AMPK levels, with the use of metformin and resveratrol, two activators of AMPK. We have previously detailed some of the molecular changes during inflammation injury following acute dextran sodium sulphate (DSS)-treatment of *Rassf1a* knockout mice [41]. The use of DSS in the *Rassf1a^−/−^* or *Rassf1a^+/−^* mice, resulted in severe inflammation injury, with <20% survival of the mice (Figure 3C, left panel). When metformin is fed to the *Rassf1a^+/−^* mice at 2 g/L, we observed a dramatic recovery from inflammation injury (Figure 3C, left panel), with a regain of normal levels of active AMPK in both lysates and IHC sections from the colons of metformin-treated animals (Figure 3C, right panel). This can also be observed for treatment utilizing another AMPK activator, resveratrol, in the food pellet, during DSS induced inflammation injury (Appendix A). Similar to the use of metformin, we can observe a regain of activation status of AMPK in colon lysates of resveratrol-treated animals (Appendix A). Interestingly, in the spontaneous model of IBD, the *Il-10^−/−^* mice, we observed a loss of active AMPK upon DSS treatment (Appendix A), suggesting a role for AMPK and proper regulation of metabolic homoeostasis in IBD. The metabolic reset provided by metformin or resveratrol, appears to be sufficient to allow for >80% survival of these normally DSS-sensitive animals. Although human IBD is far more complex than normally observed in the DSS model, the result with the use of 2 g/L of metformin may provide a framework for a human trial for the use of metformin to treat or manage acute to severe IBD.

### 2.4. The Activation Status of Proliferation Driver and Transcription Factor Yes Associated Protein (YAP) Is Elevated in IBD Patients

YAP (known as Yorkie [Yki] in Drosophila) is a key driver of proliferation, linked to the TEAD family of transcription factors, and an end effector of the Hippo pathway [73]. Removal of either Yki from intestinal stem cells in Drosophila or YAP in *Yap^−/−^* mice, revealed poor survival and decreased epithelial cell proliferation in response to DSS treatment (characteristics similar to what we have observed in DSS-treated *Rassf1a^−/−^*) [74,75]. RASSFA is an upstream modulator of the Hippo pathway [76] and there has been documented observations linking AMPK to YAP biology. We have observed previously that in the DSS treated *Rassf1a^−/−^* knockout model we can detect a robust increase in active YAP (YAP pY357 levels), suggesting abnormal YAP transcriptional activity upon inflammation injury [41]. In this study, we explored the proliferative status of YAP in human IBD colon sections and observed substantial detection, in both CD and UC patients, of YAP pY357 and YAP pS94, two activation states of YAP (Figure 4A,B). Both activation states of YAP have been demonstrated to drive proliferation in numerous cells [77]. We suspect that this activity is driven by abnormal inflammation and increased proliferation, that may be an interesting predictor of abnormal growth/malignancy if inflammation is not controlled.

### 2.5. Correlations between the Biomarkers Explored in This Study

Disease prediction is extremely difficult unless robust biomarkers are identified, that truly reflect the molecular changes that may occur. For IBD, there are no reliable biomarkers to date, mainly because the focus has been on inflammatory drivers. Thus far, we have evidence that inflammatory drivers, metabolic, and proliferative markers all intersect to drive IBD development. We have identified the tumor suppressor gene RASSF1A; the obligate kinase of the intracellular pattern recognition NOD2 receptor RIPK2; the metabolic regulator AMPK; and the proliferative driver YAP, as possible molecular drivers of IBD. We carried out correlation analyses to better understand the relationships between these markers. Figure 5A reveals significant correlations between active RIPK2 and active AMPK and active YAP; significant correlations between RASSF1A expression and active RIPK2, active YAP, and active AMPK, suggesting an interplay of these four molecular drivers of inflammation, leading to metabolic and proliferative changes in the colon of IBD patients.

Furthermore, Figure 5B illustrates that the changes in RIPK2, YAP, AMPK, and RASSF1A appear more pronounced in patients with a long-standing disease, of >10 years. Interestingly, with respect to the activation of RIPK2, standard IBD drug treatment does not appear to resolve the issue of abnormal RIPK2 activation (Figure 5C), nor does clinical remission reflect changes in RIPK2 activity (Figure 5D). In addition, there appears to be weak or no correlation between active RIPK2 and changes in c-reactive protein (CRP) or faecal calprotectin (Appendix A). RIPK2 is a molecular driver of inflammation, and Figure 5C,D suggest that, although some patients may be in clinical remission, their RIPK2 status should be monitored, before levels escalate to those found in patients with long-standing disease.

### 2.6. Correlations between Leukocytes and Matched Biopsies from IBD Patients

Detection of biomarkers is limited to accessibility of biological material, in order to analyze the biomarker. We explored the possibility of detecting the activation status of some of the biomarkers identified thus far, in the blood of IBD patients. We looked at several matched blood/biopsy samples, and found identical patterning of changes between non-IBD and IBD patients. We had robust detection of RIPK pY474, loss of both AMPK α1 T172 and RASSF1A (Figure 6), and robust detection of YAP pY357 in leukocytes (Figure 6D). Interestingly, for both RIPK2 the AMPK, the changes appear to be equal or more robust in the leukocyte fraction. These data suggest that a non-invasive option could be developed, to monitor the status of RIPK2, AMPK, and YAP. 

### 2.7. RASSF1A, RIPK2, and YAP Have Robust Changes in Patients That Have IBD and Progress to CRC

Genetically and epigenetically, we are just beginning to uncover the susceptibility loci between IBD and CRC [78]. In general, abnormal molecular pathways in IBD (both CD and UC) link to regulators of inflammation, autophagy, and cell death pathways, whereas abnormal molecular pathways of CRC link to growth control and proliferative signaling pathways. With access to biopsies and tissue blocks at different time points from individual case studies (patients A–D that progressed from IBD to CRC), we explored the expression status of RASS1A, RIPK2, AMPK, and YAP, and studied whether more profound changes were observed when CRC was compared to UC with no CRC. Figure 7 (left panel) and Appendix A shows the snapshot of biomarker staining during no disease, UC, and CRC stages, as determined by endoscopy and pathological analyses. The right panel is a quantitative summary of numerous sections in each category for five UC-CRC patients, illustrating the changes that occurred in these biomarkers as the patient progressed from no disease, to UC, to CRC. What is clearly evident, is that (a) during clinical remission, these biomarkers are still elevated and may be causing low level inflammation that may trigger relapses an/or CRC progression; and (b) the changes observed for each biomarker appear to be more robust in UC patients that progress to CRC than in UC patients with no CRC. We believe biomarker levels during UC may be utilized to better monitor patients that are predisposed to developing CRC. Further analysis is required in a larger dataset.

## 3. Discussion and Conclusions

IBD includes Crohn’s disease and ulcerative colitis, both of which are highly prevalent [79,80]. Current treatment is extensive and often requires lifelong immunotherapy. The majority of adult treatments involve biologics such as anti-TNFα antibodies (infliximab/remicade) and steroidal products. However, 50% of patients lose response to infliximab, and patients cannot be on prolonged use of steroids. Several other therapies include aminosalicylates, corticosteroids, and disease-modifying anti-rheumatic drugs (azathioprine, cyclosporine, methotrexate), to mention a few [81,82,83]. Those with long-standing colitis (UC > 10 years) are at increased risk for CRC requiring invasive colonoscopies every 1–2 years, as screening biomarkers have not been identified.

In this study, we have explored not just the inflammatory drivers of IBD but also the epigenetic, metabolic, and proliferative drivers. RASSF1A is one of the most epigenetically silenced tumor suppressors in cancers, that restricts NFκB activity [41]. We have demonstrated the importance of RASSF1A and YAP during acute inflammation [41] and have evidence of a role of both RASSF1A and YAP in driving chronic inflammation and malignant transformation (unpublished observations). We have also demonstrated in inflammatory breast cancer (IBC), that there is a correlation between loss of expression of RASSF1A and activation of RIPK2, as indicated by increased detection with the RIPK2 pY474 phosphospecific antibody [53]. Similar to what we published for IBC and CRC, RASSF1A is epigenetically silenced in IBD (Figure 1), with an identical epigenetic signature to CRC, the “IBD-CRC epigenetic signature”, suggesting that IBD is epigenetically a predisposing factor to CRC. The consequence of the loss of RASSF1A expression, is elevated phosphorylation of RIPK2 on Y474 (and thus activation of RIPK2, Figure 2), loss of AMPK activity as measured by detection levels of (AMPKα1) T172 (Figure 3), and increased YAP activity (as measured by both pY357 and pS94 phospho-levels, Figure 4). 

We can observe a very good correlation between RASSF1A expression and active RIPK2 and active AMPK (Figure 5). We have evidence that RASSF1A can physically restrict association of RIPK2 with NOD2, to prevent activation of RIPK2 (Said et al., manuscript in preparation), to explain a molecular link between epigenetic levels of RASSF1A and the activation of the NOD2/RIPK2 pathogen pathway. Lastly, in breast cancer and a few other cancers, AMPK is thought to modulate the activation status of YAP, to control it proliferative capacity [84]. Thus, in the absence of RASSFA, RIPK2 activation is uncontrolled and thus contributes significantly to the inflammatory response. The loss of RASSF1A also results in an altered metabolic state, with a loss of AMPK activity, loss of colonic insulin production, and an increased YAP proliferative capacity, as observed in patient sections and in our DSS-induced inflammation model (Figure 3, Appendix A). We speculate that abnormal RIPK2 activity during acute to chronic inflammation episodes, may promote stromal inflammation, proliferation, and altered metabolism, leading to malignant transformation.

Furthermore, the finding that the loss of AMPK activity, coupled to the loss of insulin production in the colon, identifies two molecular drivers of metabolic syndrome disorder in IBD patients. Loss of AMPK is robustly lost in CD and UC patients, as is insulin production. In this study, we demonstrated that, when a reset of metabolism occurs with metformin or resveratrol, active RIPK2 is significantly inhibited (Figure 3C and Appendix A), AMPK levels are restored, and YAP S94 activity (and to a large extent YAP Y357 activity) are inhibited, suggesting that a metabolic reset with metformin or resveratrol treatment can alleviate RIPK2-driven inflammation (Appendix A). When a RIPK2 inhibitor is administered to our mouse model of inflammation, we can observe a regain of AMPK levels and significant loss of YAP activity (Appendix A), suggesting that inhibiting RIPK2 can also lead to a metabolic reset and may promote remission in IBD patients. Thus, one can suggest the use of both metformin and RIPK2 inhibitors as treatment schemes for IBD, depending on the activation status of RIPK2 or AMPK (please see Figure 8). Our RIPK2 inhibitor was extensively characterized for inhibition of RIPK2 [85], and we have evidence that the tumor suppressor RASSF1A, can physically associate with RIPK2, and keeps in check its ability to drive inflammation (Said et al., manuscript in preparation).

There are many cellular consequences to the loss of AMPK activation, that will need to be explored further in IBD patients. It is thought that AMPK can inhibit the growth of cancer cells by switching off protein synthesis and cell proliferation [62,86]. Several epidemiological studies have suggested that diabetes is associated with an increased risk of certain cancers [59]. Indeed, hyperglycemia and hyperinsulinemia are thought to promote the growth of cancer cells in diabetic patients [87]. Metformin, the most commonly prescribed oral anti-diabetic medication and an activator of AMPK, has been shown to have strong anti-proliferative and/or pro-apoptotic properties in several cancer cell lines, and may be of benefit to diabetic cancer patients [88]. Although controversial, the anti-cancer effect may be independent of AMPK activation [89]. Interestingly, several studies have shown that the use of metformin is associated with lower risk of colon and pancreatic cancer in type 2 diabetic (T2D) patients [90]. We performed a database search for the prevalence of IBD in insulin and metformin users in Alberta, and preliminary analysis indicates that IBD is prevalent in approximately 1 in 300 insulin mono-users and 1 in 900 metformin mono-users, when compared to 1 in 150 non-users of insulin or metformin (unpublished information). These data are in line with the results of Tseng (2020), who compared the risk of IBD between users and non-users of metformin [91], and others that suggest a role for metformin in managing inflammation [92]. The results of our database analysis are also in line with our animal model results, that clearly show that metformin intake was sufficient to reverse the damaging effects of DSS-induced inflammation injury (a colitis model) in the inflammation sensitive *Rassf1a^−/−^* mouse. Thus, insulin, and more importantly metformin, protected against the occurrence of IBD in Alberta. Serum/colon tissue AMPK levels will help in deciding if metformin may be a useful drug as a primary or a co-treatment for IBD.

The most interesting aspect of our study was the detection of biomarker changes between biopsy and blood (leukocytes). In a small subset of IBD patients, with disease > 10 years, we can observe matched changes in active RIPK2, active AMPK, RASSF1A, and active YAP (Figure 6). Interestingly, there was a more robust difference in RIPK2 status in the leukocyte fraction, suggesting further exploration of the use of an active RIPK2 biomarker status in determining therapy selection, as modeled in Figure 8. The most important markers may be RIPK2 and YAP, to indicate inflammation and the beginning signs of hyperproliferation, respectively, and possible progression to CRC. 

The biomarkers identified in this manuscript may be useful in understanding the origins of acute and chronic intestinal inflammation, what sustains it, and how it promotes the malignant state if uncontrolled. There is a need to better understand the inflammatory mediators/molecular drivers involved in promoting the pathogenesis of CRC, and the biomarkers identified in this study may be essential starting points to a better understanding of the origins of malignant transformation. Eliminating inflammation through novel modulation of metabolism, proliferation, and epigenetics, will influence the composition of the microbiome and the metabolic and inflammatory state of the colonic microenvironment, and significantly contribute to rational drug design. Proposing novel combination therapies will not only help IBD patients but may aid in CRC prediction and reduce the incidence of CRC. We propose that controlling intestinal inflammation is an important cancer prevention strategy, that would decrease the incidences of CRC and possible other inflammatory driven cancers. The biomarkers identified in this study will help in identifying at risk populations of long-standing IBD in a non-invasive manner, or at least warrant surveillance colonoscopies when needed. IBD and IBD-CRC require a multidisciplinary approach, to identify novel targets for disease prevention (or screening), to offer novel, precision therapeutics, that will reduce disease symptoms and unnecessary costs and procedures. 

## 4. Materials and Methods

### 4.1. Patient Collection

We have collaborated with researchers in the Centre for Excellence in Gastrointestinal Inflammation and Immunity Research (CEGIIR), at the University of Alberta, the Alberta IBD Consortium, and the Alberta Health Services pathology bank, to obtain patient biopsy samples as necessary. Samples were collected from patients with active disease, having gross inflammation, and from those in remission from areas showing no gross inflammation, when available. Biomarker analysis was conducted on a subset of these patients, to time constraints. This was carried out under research ethics protocol # Pro00077868.

We also received blood and tumor biopsy samples from colorectal cancer patients, through collaborations with the Cross Cancer Institute Tumor Bank and Dr. Oliver Bathe (University of Calgary). All biopsy and tumor samples were flash-frozen in liquid nitrogen and stored at −80 °C. Samples were transferred to Z-Fix fixative for 24–36 h in order to fix the samples, prior to being dehydrated and paraffin-embedded, as described previously [41]. Biopsy samples were also mounted for IHC immunoblotting, to assess the usefulness of novel IBD biomarkers identified in the mouse models. This was carried out under research ethics protocol # Pro00077868.

### 4.2. Patient Chart Review

Retrospective chart reviews were completed for all case study patients and the majority of the IBD patients. The inclusion criteria were as follows: newly diagnosed patients (<5 years) and patients with long-standing IBD disease (>10 years). Patients with other co-morbidities such as diabetes, all cancers, celiac disease, or irritable bowel syndrome were excluded. Tissue samples were obtained for immunoblotting and immunohistochemistry, as indicated, in addition to blood samples for leukocyte analysis. This was carried out under research ethics protocol # Pro00001523. Demographics are indicated in Table 1 and Table 2 in results Section 2.1.

### 4.3. Acute Mouse Model of Inflammation

Intestinal colitis can be modeled in mice by the addition of 3% dextran sodium sulfate (DSS, #160110, molecular weight of 10,000, MP Biomedicals, Santa Ana, CA, USA) in drinking water for 7 days (acute treatment), to induce injury, followed by regular drinking water for recovery (as described previously) [20]. DSS irritates the colonic mucosa, resulting in epithelial wall breakdown, microflora invasion activating TLR-expressing epithelial cells, and mucosal injury. Mice were monitored over time for weight changes, as well as clinical symptoms of colitis such as rectal bleeding and diarrhea. Animals were euthanized once rectal bleeding became grossly apparent. Animals were sacrificed at various points, and blood, colon, kidney, and liver samples collected for molecular analysis. The animal ethics research of the University of Alberta in Edmonton, AB, Canada approved the study (under protocol numbers #461 and 639).

### 4.4. Tissue Collection, Preparation and Pathology Scoring and Colon Lysates

Colon samples were isolated, fixed in Z-Fix (Anatech 170) and paraffin-embedded. All inflammation scores were obtained utilizing blinded scoring by a gastrointestinal pathologist (Dr. Aducio Thiesen), based on infiltration of enterocytes, neutrophils, lamina propria cellularity, crypt structure, and epithelial hyperplasia (scored as 0–2, where 2 = maximal injury) [41]. For colon lysates, samples intended for protein analysis were flushed with 1X PBS, to clear fecal matter, and immediately placed in ice-cold T-PER lysis buffer (Thermo Fisher Scientific, Waltham, MA, USA) with fresh aprotinin (0.1%), phenylmethylsulfonyl fluoride (PMSF) (0.2%), sodium pyrophosphate (NaPP) (0.1%), sodium orthovanadate (1%), and a protease inhibitor cocktail. Samples were then homogenized using a Fisher PowerGen handheld homogenizer, and centrifuged at 4 °C, at max speed, for 10 min. The supernatants, containing proteins, were stored at −80 °C for further use. Protein concentration was determined using the Bradford protein assay.

### 4.5. Immunohistochemistry

Colon samples from selected patients were fixed in formaldehyde, paraffin embedded, and mounted. Samples were rehydrated and antigen retrieval performed using boiling sodium citrate. Immunodetection of proteins of interest was carried out using 1:100 dilution of antibodies against the desired proteins, followed by signal amplification, using a biotin-labelled secondary antibody, streptavidin-HRP amplification, and final detection using metal-enhanced 3, 3′-diaminobenzidine (DAB).

### 4.6. Immunoblotting

Polyacrylamide mini gels, 5% stacking and 7.5% separating, were prepared as published previously [41].

### 4.7. Immunoblotting Leukocyte Fractions

Patient peripheral blood was drawn into an EDTA-containing tube (about 5–10 mL) and red cell lysis buffer (bicarbonate-buffered ammonium chloride solution: 0.826% NH_4_Cl, 0.1% KHCO_3_, 0.0037% Na_4_EDTA in H_2_O) was added, at a ratio of 20:1 (lysis buffer/blood), and incubated for 10 min. Once the erythrocyte fraction was lysed, the samples were centrifuged for 10 min at 400× *g* or 3500 rpm, using a tissue culture centrifuge. The leukocyte pellet was washed twice with 1X PBS before being divided into three fractions: one fraction, containing ½ sample, was used for immunoblotting. Cells were lysed in lysis buffer containing 50 mM HEPES (pH 7.5), 150 mM NaCl, 1 mM MgCl_2_, 1.5 mM EDTA, 0.5% Triton X-100, 20 mM β-glycerolphosphate, 100 mM NaF, and 0.1 mM PMSF. Total lysates were assayed (Bradford) for protein concentration and loaded, after boiling (7 min), in 4X SDS-PAGE Sample Buffer, such that approximately 40 µg were loaded onto 10% SDS-polyacrylamide gels, run, transferred and immunoblotted as previously published [20]. Blocking was done in 5% BSA (in TBS-T) and primary/secondary antibody diluted in 5% BSA (in TBS-T). 

### 4.8. DNA Methylation Analysis by Pyrosequencing

Pyrosequencing, to determine methylation status, was carried out using the Qiagen Pyromark Advanced kit, according to the manufacturer’s instructions, as previously carried out [68]. Briefly, genomic DNA isolated from mouse colonic tissue, was bisulfite modified using the Qiagen Epitect Bisulfite Conversion kit, using the instructions for “sodium bisulfite conversion of unmethylated cytosines in DNA from low-concentration solutions”, according to the manufacturer’s instructions. The resulting bisulfite-modified DNA was then used as a template to amplify the region of interest, using a biotinylated primer set for the first exon of Rassf1a, provided by Qiagen (cat # PM00416290). The Assay1 covers 11 CpGs in promoter and 1 CpG in exon1 of the RASSF1A. The Assay2 covers 20 CpG (Cytosine-phosphate-Guanine) located right upstream of the 12 CpGs covered by the Assay1 [41]. The PCR was performed using a PyroMark PCR Kit (Qiagen, Germantown, MD, USA), in a volume of 25 µL, containing 12.5 µL of 2× PyroMark PCR Master Mix, 1.25 µL of each PCR primer (5 µM), 2.5 µL of 10× Coral Load Concentrate, 6.5 µL high purity water, and 1 µL of bisulfite-treated template DNA. The PCR cycling programme for both primer sets was composed of an initial Taq activation/DNA denaturation step at 95 °C for 15 min, followed by 50 cycles of denaturation at 95 °C for 30 s, annealing at 58 °C for 30 s, and elongation at 72 °C for 30 s. The program was finished by a final elongation step at 72 °C for 10 min. The PCR product (7 µL) was visualized by gel electrophoresis, and 10 µL was subjected to the sample preparation process for pyrosequencing. The sequencing results were analyzed using the Advanced PyroMark software (Qiagen). A control PCR reaction, without template DNA (non-template control), was included in the assay. Pyromark assays were carried out two times, for accuracy.

### 4.9. Myloperoxidase Activity Analysis

All activity assays were performed in duplicate or triplicate, on 96-well microtiter plates, and analyzed with a microplate reader. Peroxidase activity, with 3,3′,5,5′-tetramethylbenzidine (TMB, Sigma, Oakville, ON, Canada), was measured as described previously [26]. Briefly, 10 μL of sample was combined with 80 μL 0.75 mM H_2_O_2_ (Sigma) and 110 μL TMB solution (2.9 mM TMB in 14.5% DMSO (Sigma), and 150 mM sodium phosphate buffer at pH 5.4), and the plate was incubated at 37 °C for 5 min. The reaction was stopped by adding 50 μL of 2 M H_2_SO_4_ (Sigma), and absorption was measured at 450 nm, to estimate MPO activity [41,93].

### 4.10. Antibodies

The following antibodies were utilized for this study: rabbit anti-RIPK2 (Santa Cruz sc-22763), rabbit anti-pY 474 RIPK2 (in-house made), rabbit anti-ERK1 (Santa Cruz sc-93) and rabbit anti-ERK2 (Santa Cruz sc-154), rabbit anti-(AMPKα1) pT172 (Cell Signalling #2535S), mouse anti-(AMPKα1) (Cell Signalling #3532), rabbit anti-YAP Y357 (Phospho-YAP (Tyr357), Sigma-Aldrich Y4645), rabbit anti-YAP pS94 (a generous gift of Dr. Marius Sudol), and mouse anti-RASSF1A monoclonal antibody (in-house made). All in-house antibodies were validated using purified proteins or positive and negative cells. Rabbit anti-pY 474 RIPK2 can now be purchased from QED Biosciences (San Diego, CA, USA), that documents the characterization of rabbit anti-pY 474 RIPK2. Mouse anti-RASSF1A was characterized by ELISA analysis using RASSF1A gel slices, and only high-titer clones were selected and purified. 

### 4.11. Statistical Analyses

Statistical analyses were performed using one-way or two-way ANOVA with Tukey or Bonferroni post hoc tests, respectively, or Students *t*-test (two-tailed), as indicated using the GraphPad Prism 5 software. All statistics were non-parametric. Results are considered significant if the *p*-value is <0.05. All experiments were carried out at least three times with biological replicates. Error bars in all graphs represent the standard error. For all data analysis, samples were taken from UC and CD patients at inclusion criteria of being diagnosed as IBD, with no sex bias or age bias, for this study. These inclusion criteria were carried out before statistical analysis was performed. The number of samples was determined using the sample size calculation: n = Z^2^ [*p*(1 − *p*)]/(D^2^) and within the 90–95% confidence interval. Z equals the z-score [1.65 for 90% confidence interval and 1.96 for 95% confidence interval), *p* is the standard deviation, and D is the margin of error. Using this power calculation, n = 106–150 or greater, within the 90–95% confidence interval [94].

## Figures and Tables

**Figure 1 ijms-24-05967-f001:**
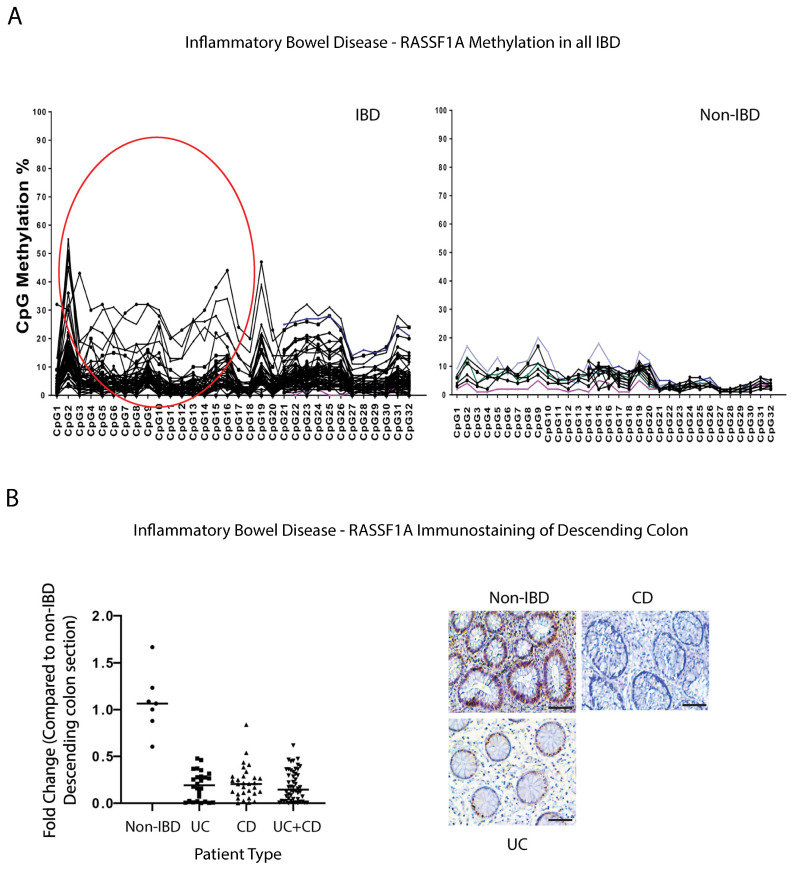
RASSF1A is epigenetically silenced in inflammatory bowel disease (IBD). (**A**) The methylation percentage of the individual CpGs in the RASSF1A promoter in inflammatory bowel disease patients, n = 80, and 14 control blood samples. The individual CpG methylation percentage indicates what percentage of DNA molecules are methylated at this site in the sample. Circled region corresponds to a hot spot in CRC patients [42]. (**B**) Immunohistochemical staining for RASSF1A in descending colon sections of IBD patients reveals loss of expression in IBD, confirming methylation data in (**A**). Left panel, summary of fold change in IHC staining of tissue sections; right panel, representative sections for each patient category. *p* value < 0.001 for difference between UC/CD/UC+CD vs. non-IBD (n = 10 for non-IBD and 26 for UC, 32 for CD, and 58 for UC+CD).

**Figure 2 ijms-24-05967-f002:**
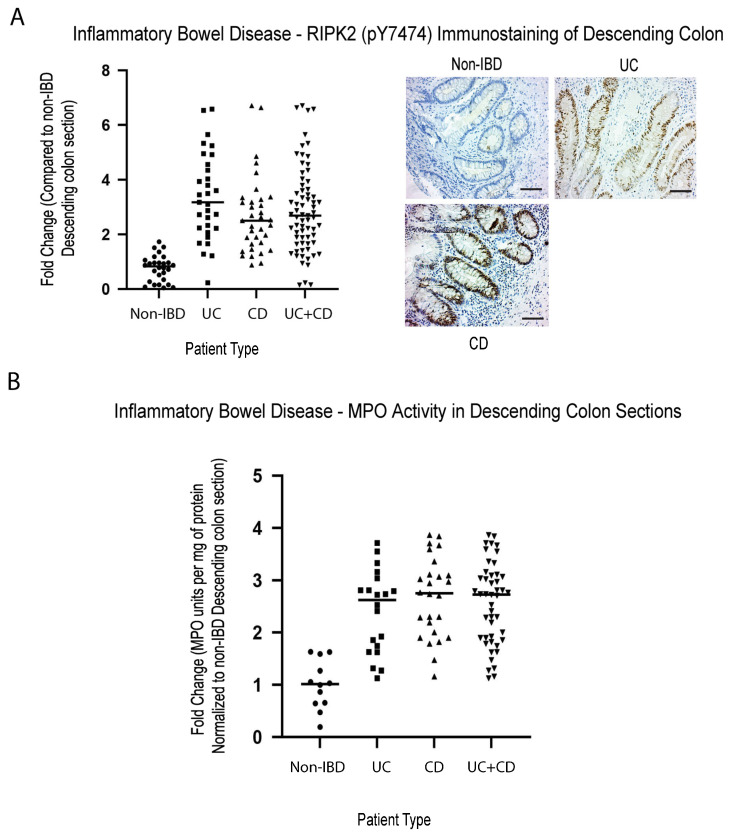
Elevated activity of inflammatory markers (active RIPK2 and MPO) in IBD colonic sections. Immunohistochemical staining for (**A**) RIPK2 pY474, (**B**) myeloperoxidase in descending colon sections of IBD patients reveals elevated activity of the pathogen recognition receptor (NOD2) associated kinase RIPK2, and a common molecular marker of inflammation, MPO. For (**A**), left panel, summary of fold change in IHC staining of tissue sections; right panel, representative sections for each patient category. For (**A**), *p* value < 0.001 for difference between UC/CD/UC+CD vs. non-IBD (n = 30 for non-IBD and 32 for UC, 35 for CD, and 67 for UC+CD). For (**B**), *p* value < 0.001 for difference between UC/CD/UC+CD vs. non-IBD (n = 12 for non-IBD and 21 for UC, 29 for CD, and 50 for UC+CD).

**Figure 3 ijms-24-05967-f003:**
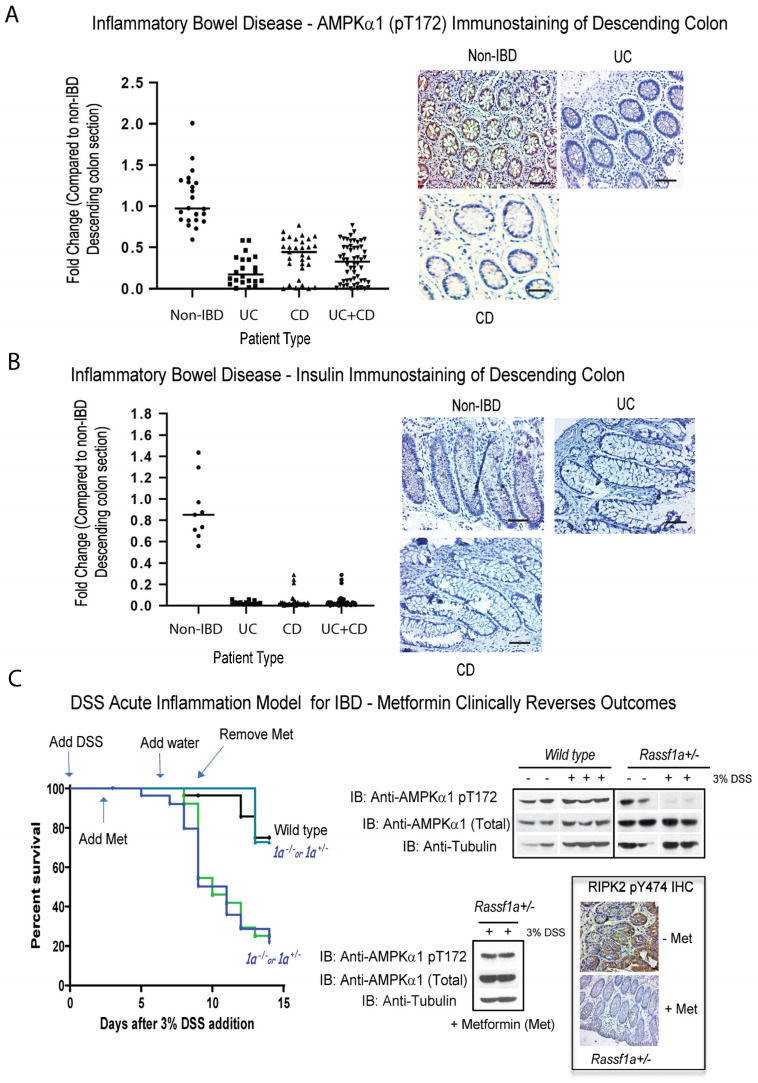
Loss of metabolic markers in IBD and a metabolism-directed therapeutic intervention in an IBD mouse model. Immunohistochemical staining for (**A**) (AMPKα1) pT172 and (**B**) insulin in descending colon sections of IBD patients, reveal loss of activation of (AMPKα1) and tissue expression of insulin. Left panel, summary of fold change in IHC staining of tissue sections; right panel, representative sections for each patient category. (**C**) DSS-induced inflammation injury model of acute inflammation was carried out in the DSS-susceptible model, *Rassf1a^+/−^* (*1a^+/−^*). Metformin was given in the drinking water at 2 g/L (human equivalent dose [HED] = 500 mg/day) from day 3 to day 9 (during peak of inflammation injury). Mice were monitored for piloerection, bloatednesss, tremors, lack of movement, and rectal bleeding. Left panel, Kaplan–Meier curve with n = 13 (DSS/metformin treated) and n = 28 (DSS treated) animals were used for all conditions and *p*-value between wild type and *Rassf1a^−/−^.* was <0.005. Right panel, immunoblot of descending colon lysates from experiment in the left panel, with the indicated antibodies, as well as representative colon sections stained for active RIPK2 (using the pY 474 RIPK2 antibody) upon metformin treatment. For (**A**), *p* value < 0.001 for difference between UC/CD/UC+CD vs. non-IBD (n = 30 for non-IBD and 26 for UC, 34 for CD, and 60 for UC+CD). For (**B**), *p* value < 0.001 for difference between UC/CD/UC+CD vs. non-IBD (n = 14 for non-IBD and 14 for UC, 36 for CD, and 50 for UC+CD).

**Figure 4 ijms-24-05967-f004:**
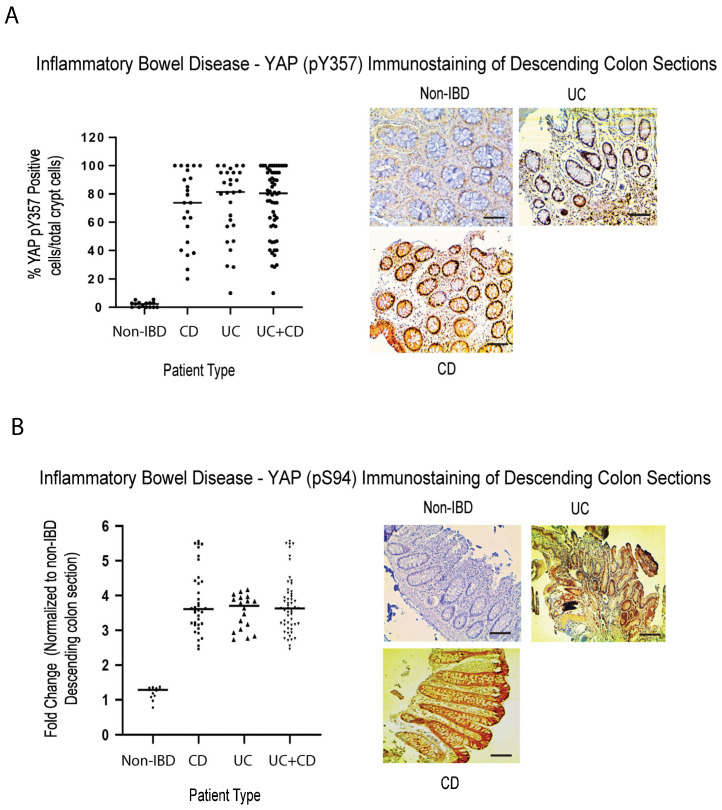
Elevated activity of active YAP in colonic sections from IBD patients. Immunohistochemical staining for (**A**) active YAP pY357 and (**B**) active YAP pS94, in descending colon sections of IBD patients, confirm elevation in YAP activity. Left panels in (**A**,**B**), summary of fold change in IHC staining of tissue sections; right panel, representative sections for each patient category. For (**A**), *p* value < 0.001 for difference between UC/CD/UC+CD vs. non-IBD (n = 15 for non-IBD and 25 for UC, 32 for CD, and 57 for UC+CD). For (**B**), *p* value < 0.001 for difference between UC/CD/UC+CD vs. non-IBD (n = 15 for non-IBD and 19 for UC, 35 for CD, and 54 for UC+CD).

**Figure 5 ijms-24-05967-f005:**
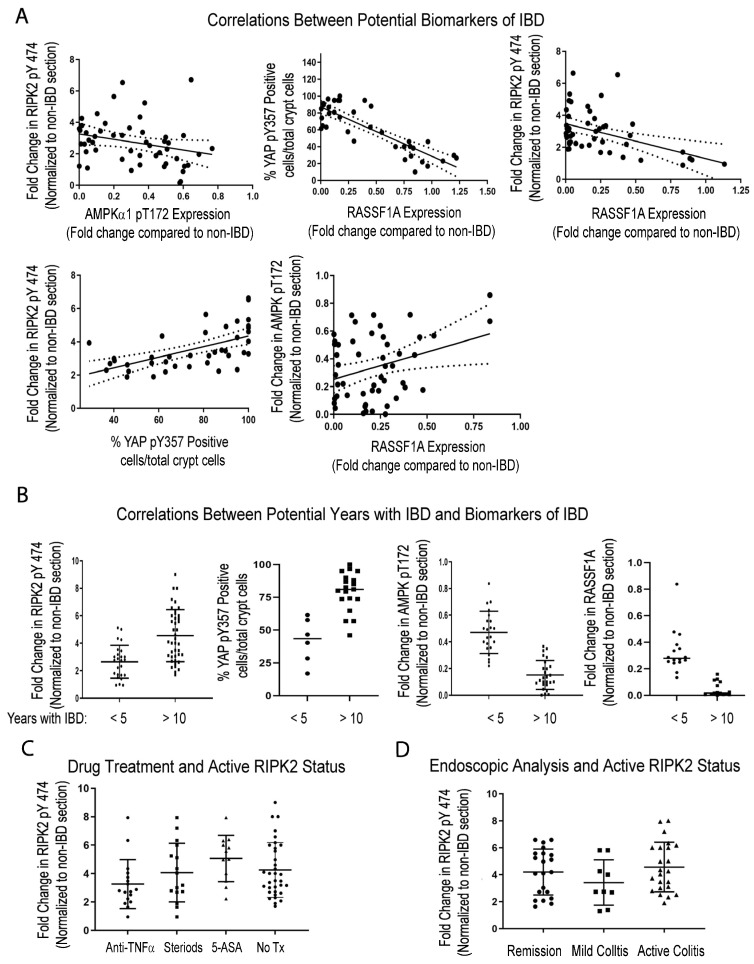
Potential correlations between four identified biomarkers of IBD—RASSF1A, (AMPKα1), YAP, and RIPK2. (**A**) Correlations between biomarkers. For top left, r^2^ = 0.07 and *p* value 0.05, n = 60–74; top middle panel, r^2^ = 0.70 and *p* value < 0.0001, n = 45–50; top right, r^2^ = 0.60 and *p* value < 0.001, n = 55–60; bottom left, r^2^ = 0.33 and *p* value 0.0001, n = 45–60; bottom right, r^2^ = 0.10 and *p* value 0.022, n = 60. (**B**) Correlations between biomarkers and duration of disease. For (**B**), *p* value < 0.001 for all comparisons explored (n = 30–40 for RIPK and (AMPKα1); n = 15–20 for YAP and RASSF1A analysis). (**C**) Correlations between active RIPK2 pY474 and therapy, before biomarker was established. For (**C**), *p* value > 0.05 for all treatments, compared to No Tx and n = 12–17 patients per drug group and 40 for the No Tx group (all patients with disease for more than 10 years). (**D**) Correlations between active RIPK2 pY474 and histologic staining of tissues during clinical management. For (**D**), *p* value > 0.05 for comparing RIPK2 pY474 fold change in patients during remission, when compared to patients with moderate to active disease (n = 10–23 in each category and most patients with disease for more than 10 years).

**Figure 6 ijms-24-05967-f006:**
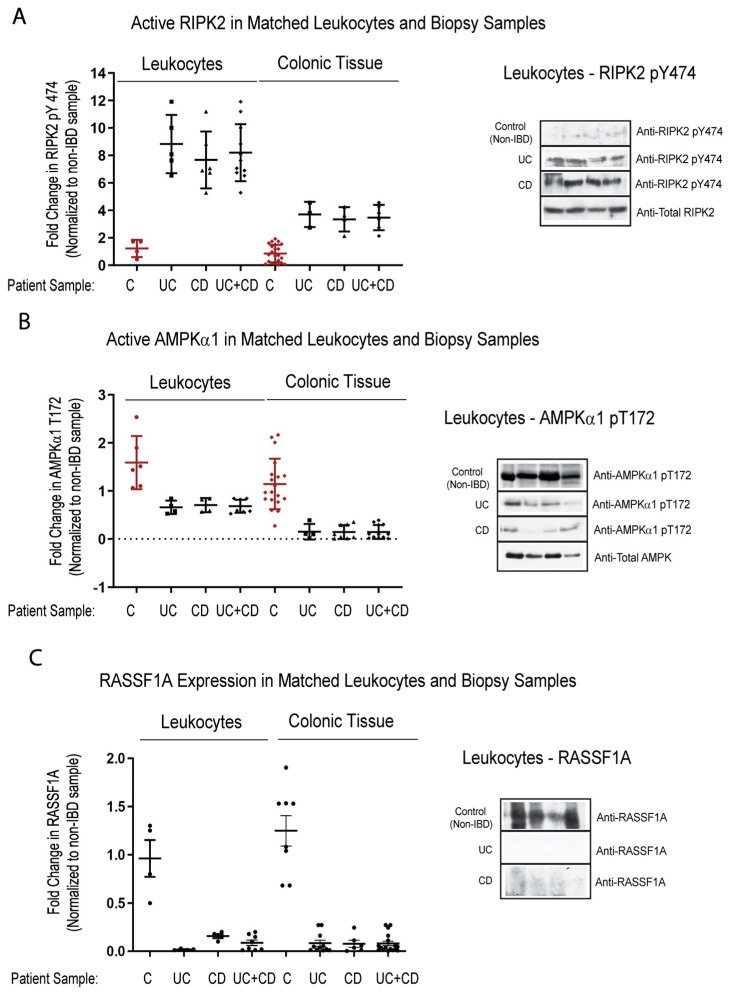
Analysis of biomarkers in matched leukocytes and descending colon sections reveal identical trends, suggesting non-invasive analysis. Leukocyte fraction and descending colon sections were obtained from non-IBD and IBD individuals as indicated. These samples were then immunoblotted for RIPK2 pY474 (**A**), (AMPKα1) pT172 (**B**), RASSF1A (**C**), and YAP pY357 (**D**). Left panels, summary of fold change in IHC staining of tissue sections and densitometrically scanned immunoblotted results; right panel, representative sections for each patient category. For (**A**) (leukocytes), *p* value is <0.001 for CD or UC or CD+UC vs. non-IBD samples (n = 5–8 for non-IBD, UC or CD, and 12 for UC+CD). For (**A**) (colonic tissue), *p* value is <0.0001 for CD or UC or CD+UC vs. non-IBD samples (n = 24 for non-IBD, n = 4 for UC or CD, and 8 for UC+CD). For (**B**) (leukocytes), *p* value is <0.02 for CD or UC or CD+UC vs. non-IBD samples (n = 6 for non-IBD, UC or CD, and 12 for UC+CD). For (**B**) (colonic tissue), *p* value is <0.001 for CD or UC or CD+UC vs. non-IBD samples (n = 20 for non-IBD, n = 4–8 for UC or CD, and 12 for UC+CD). For (**C**) (leukocytes), *p* value is <0.004 for CD or UC or CD+UC vs. non-IBD samples (n = 11 for non-IBD, 15 for UC, 7 for CD, and 22 for UC+CD). For (**C**) (colonic tissue), *p* value is <0.0001 for CD or UC or CD+UC vs. non-IBD samples (n = 8 for non-IBD, 11 for UC, 7 for CD, and 18 for UC+CD). For (**D**) (leukocytes), *p* value is <0.004 for CD or UC or CD+UC vs. non-IBD samples (n = 4–5 for non-IBD, UC or CD, and 9 for UC+CD). For (**D**) (colonic tissue), *p* value is <0.0001 for CD or UC or CD+UC vs. non-IBD samples (n = 4 for non-IBD, n = 6–8 for UC or CD, and 13 for UC+CD).

**Figure 7 ijms-24-05967-f007:**
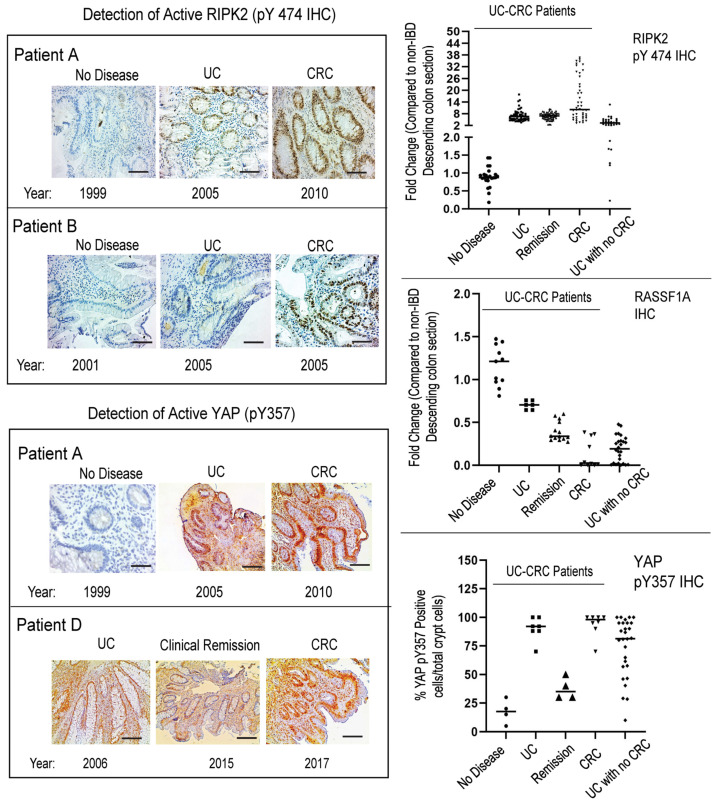
Immunohistochemical staining for RASSF1A, RIPK2 pY474, and YAP pY357 in five patients with UC-CRC. Included in this panel is stage of disease and year diagnosed. Patients B and D have since passed away, of CRC-related disease. (**Left panel**), representative sections for each category are shown; (**Right panel**), summary of fold changes for each biomarker explored. For RIPK2, *p* value < 0.0001 between UC and UC with no CRC, *p* value = 0.75 for difference between UC and UC in remission, and *p* value < 0.0001 between UC and CRC (n = 40–60 sections). For RASSF1A, *p* value < 0.0001 between UC and UC with no CRC, *p* value < 0.0001 for difference between UC and UC in remission, and *p* value < 0.0001 between UC and CRC (n = 10–15 sections). For YAP pY357, *p* value < 0.0001 between UC and UC with no CRC, *p* value < 0.0001 for difference between UC and UC in remission, and *p* value = 0.47 between UC and CRC (n = 4–8 sections).

**Figure 8 ijms-24-05967-f008:**
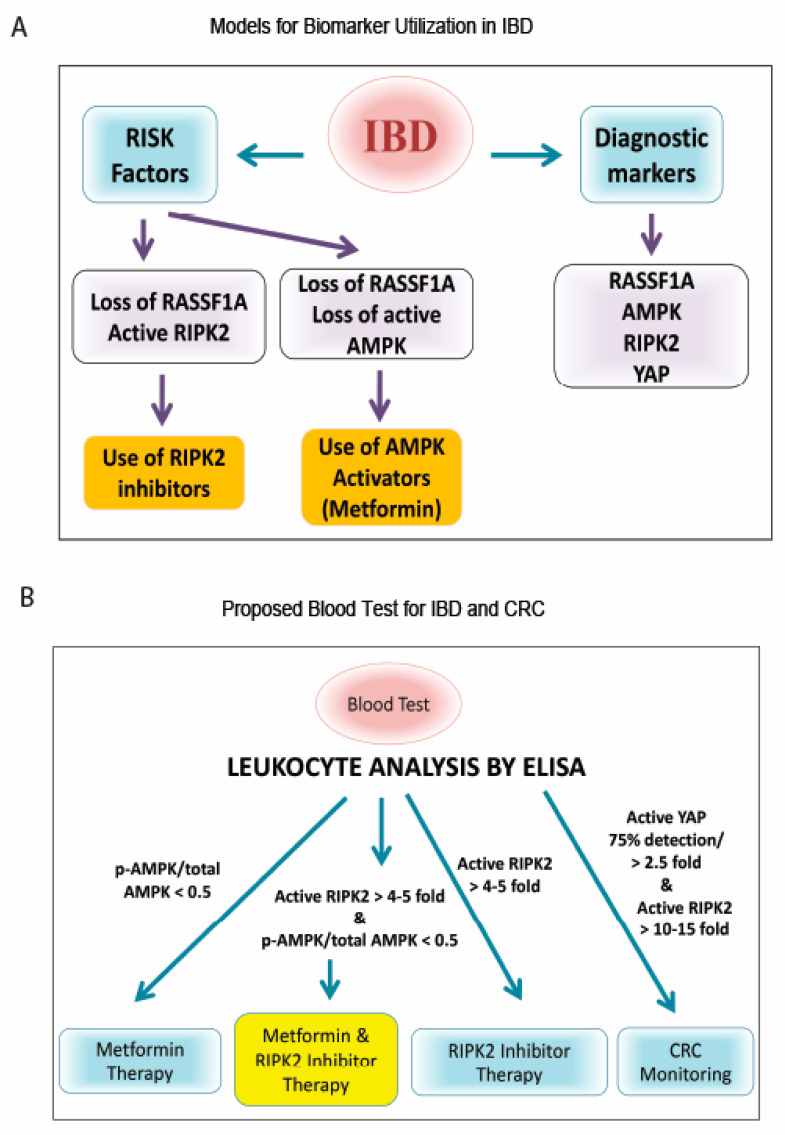
Models for biomarker utilization to monitor and treat IBD. (**A**) Flowchart of importance of RIPK2/RASSF1A/AMPK. Analysis of RASSF1A, AMPK, and RIPK2 could be used to better understand the molecular status of the patient’s colon and as a therapeutic decision making tool for personalized medicine. (**B**) Proposed blood test for IBD and CRC, based on our case study patient and Figure 7. Whole blood or leukocyte fraction can be obtained and a rapid high throughput ELISA developed, with monoclonal antibodies to ascertain activation levels. The use of an RIPK2 inhibitor and/or AMPK activator (metformin), are two therapeutic options that could benefit IBD patients, based on our study.

**Table 1 ijms-24-05967-t001:** Demographics of patients in this study. Summary of biopsies collected from patients recruited for this study. Inclusion criteria for IBD patients were CD, UC, UC-CRC, or CRC. Non-IBD excluded IBS, celiac, or GERD patients. Exclusion criteria were patients with no co-morbidities unless for UC-CRC. Of those recruited, 40–50 were used for detailed biomarker analysis, unless otherwise stated. Selection was based on yield and purity for specific assays.

Disease Type	Age, <50	Age, >50	M	F	Total Number, Biopsies	Total Number, Blood DNA
Non-IBD	59	35	60	34	94	106
UC	58	18	48	28	76	251
CD	87	43	12	118	130	303
UC-CRC	2	6	5	3	8	8
CRC	Unknown	Unknown	12	18	40	40

**Table 2 ijms-24-05967-t002:** Disease sub-types in collected biopsies. Summary of endoscopic analysis at the time biopsies were collected, based on 121 chart reviews.

IBD Subtype	Inactive	Left Sided UC	Pan-Colonic UC	Ulcerative Proctitis	Ileo-Crohn’s Disease	Crohn’s Colitis	Ileo-Colonic Crohn’s Disease	Total Analyzed
UC	18	17	24	16	NA	NA	NA	75
CD	5	NA	NA	NA	12	15	14	46

## Data Availability

The data presented in this study are available on request from the corresponding author.

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
