# Peer review of "Novel Biomarkers for Inflammatory Bowel Disease and Colorectal Cancer: An Interplay between Metabolic Dysregulation and Excessive Inflammation"

_ijms, 2023, doi:10.3390/ijms24065967_

Round 1

Reviewer 1 Report

The Authors propose in this article the importance of 4 biomarkers that were involved in tumor suppression, proliferation, inflammation and metabolism within the colon to suggest the need to find therapeutics to IBD that will reset abnormal inflammation, metabolism, proliferation and epigenetic silencing in order to drive patients into full remission. 

1.      Although I think that this data has a great scientific soundness, I don't understand why the supplementary file’s figures are of very good quality while the figures and graphs in the main a text are of very bad quality (they are all grainy).

Please the authors of revise all figures in main text and upload the best quality figures that they can. You must prepare the figures with a quality equal to that which you would charge for the final publication, so that I can compare them well with what is written in the text.

2.      The authors have to re-upload best western blot of these photos of supplementary figures:

-Figure 6A top

-Figure 6D CD isn't so clear

If they don't have better photos, is necessary replicates the experiment to have better images.

3.      Method:

a.       Why did the authors not use the cellular stratification method with the medium Ficoll-plaque TM to extract lymphocytes from peripheral blood? Maybe, because the authors also wanted the high-density neutrophils? or for other reasons? 

b.      I think that following the method described in general methods’ section 2.7, the authors also extracted the total serum proteins and platelets that were eliminated with the red blood cell lysate. Can the authors explain how they are sure that they got only the lymphocytes proteins?

c.       Expand section 2.4 of the general methods and explain well how the colon tissue samples were obtained.

d.      For section 2.10 of antibodies, how did the authors validate the antibodies developed at home and those donated by other researchers? Please authors add in the text.

e.       For section 2.11 of the statistics, add in the text how the authors verified the distribution of their data before performing the described statistical tests.

4.      Text corrections:

a.       Lane 37 change in symbol (AMPK)1

b.      Control the size of the text from the lane 66 to the lane 79

c.       Lanes 133-145-318-323 change in symbol: NFB in NFkB

d.      Lane 179 control and change the ref “376” in superscript

e.       Lane 221 add acronym of SB

f.        Lane 232 add reference of (Volodko et al, Cancers)

g.      Lane 239 add acronym of CpGs

h.      Lane 259 add superscript of reference 26

i.        Lane 320 change TLR in TRLs

j.        Lane 360 change in symbol AMPK1

k.      Lane 384 change in symbol AMPK1

l.        Lane 572 change anti-TNF

m.    Lanes 575-577 some of the text is in bold, is it for some reasons?

n.      Lane 594 change in symbol AMPK1

5.      Literature

I checked the literature and noticed that the various works are a bit dated. There are only 3 quotes from 2021 and two from 2020, the others are older. Please update the literature.

Have a good work.

Author Response

Reviewer ONE

The Authors propose in this article the importance of 4 biomarkers that were involved in tumor suppression, proliferation, inflammation and metabolism within the colon to suggest the need to find therapeutics to IBD that will reset abnormal inflammation, metabolism, proliferation and epigenetic silencing in order to drive patients into full remission. 

  1. Although I think that this data has a great scientific soundness, I don't understand why the supplementary file’s figures are of very good quality while the figures and graphs in the main a text are of very bad quality (they are all grainy).

Please the authors of revise all figures in main text and upload the best quality figures that they can. You must prepare the figures with a quality equal to that which you would charge for the final publication, so that I can compare them well with what is written in the text.

Shairaz Baksh: I thank the reviewer in commenting on our figures. I have looked at all figures at 300% viewing magnification and > 95% of the figures do not appear to be grainy. However, I have replaced the following:

Figure 1A was replaced and Figure 1A (right panel) was added to show CpG % methylation in control blood. We felt that this was missing in the original submission.

Figure 1B graph was replaced for a better resolution version as well as the left panel.

Figure 2A and B graphs were replaced for better resolution versions.

Figure 3A and B graphs were replaced for better resolution versions.

Figure 4A, top left panel was replaced for better a resolution version.

Figure 4a and b graphs were replaced for better resolution version.

I have assembled my figures in Adobe Illustrator that has 320 dpi resolution and then coverted to PDF for smaller file sizes. The final figures will be submitted as Adobe Illustrator files that will be at 320 dpi resolution. In addition all graphs were at 300 pixels/cm for better resolution.  I hope this is acceptable to the reviewer as I do not know how much more I can do to enhance resolution of images

  1. The authors have to re-upload best western blot of these photos of supplementary figures:

-Figure 6A top

-Figure 6D CD isn't so clear

If they don't have better photos, is necessary replicates the experiment to have better images.

Shairaz Baksh:  I thank the reviewer in commenting on our figures. I am uncertain what he/she is referring to as they mention Figure 6 but also mention Supplementary figures. We have zoomed to 300% and these figures appear of good quality. As mentioned previously, the final figures will be submitted as Adobe Illustrator files that will be at 320 dpi resolution. I hope this is acceptable to the reviewer as I do not know how much more I can do to enhance resolution of images. To note, CD is not clear as it does not have a signal but graininess in the lanes. This was done on purpose to show the contrast with blots from non-IBD patients that in fact does reveal a robust signal for RASSF1A in Figure 6D if that is the figure the reviewer is referring to.

  1. Method:
  2. Why did the authors not use the cellular stratification method with the medium Ficoll-plaque TMto extract lymphocytes from peripheral blood? Maybe, because the authors also wanted the high-density neutrophils? or for other reasons? 

Shairaz Baksh:  I thank the reviewer in commenting on this method. We wanted to capture the signal is all cells after red cell lysis. Thus we did not attempt to separate using a Ficoll-plaque TM  approach or any other protocols. In this way, we observed a more complete signal that may reflect RIPK2 activation in multiple immune cells. IBD, as the reviewer may know, is such a varied disease involving the activation of multiple immune cells. Future research will involved cellular stratification analysis to really answer the question of RIPK2 activation in specific immune cells. My predication is that we will still observe detection is multiple immune cells. I hope this will explain adequately explain our approach.

  1. I think that following the method described in general methods’ section 2.7, the authors also extracted the total serum proteins and platelets that were eliminated with the red blood cell lysate. Can the authors explain how they are sure that they got only the lymphocytes proteins?

Shairaz Baksh: I thank the reviewer in commenting on this method. As mentioned above, we were not only interested in the signal in lymphocytes but in other immune cells for this initial approach. Hence, we did not confirm identity of only lymphocytes in our isolated fraction. Future studies will address that question.

  1. Expand section 2.4 of the general methods and explain well how the colon tissue samples were obtained.

Shairaz Baksh: I thank the reviewer in commenting on this method. We have added this procedure to section 2.4 I hope this is acceptable for the reviewer.

  1. For section 2.10 of antibodies, how did the authors validate the antibodies developed at home and those donated by other researchers? Please authors add in the text.

Shairaz Baksh: I thank the reviewer in commenting on our in-house antibodies. We have added a description of how they were characterized. I hope this will be acceptable to the reviewer.

  1. For section 2.11 of the statistics, add in the text how the authors verified the distribution of their data before performing the described statistical tests.

Shairaz Baksh: I thank the reviewer in commenting on our in-house antibodies. We have added a description of how they were characterized, “For all data analysis, samples were taken from UC and CD patients at inclusion criteria of being diagnosed as IBD with no sex bias or age bias for this study. Only in Figure 5B do we stratify by age. These inclusion criteria were carried out before statistical analysis was done.” I hope this will be acceptable to the reviewer.” I hope this will be acceptable to the reviewer.

  1. Text corrections:
  2. Lane 37 change in symbol (AMPK)1

Shairaz Baksh: This has been corrected.

  1. Control the size of the text from the lane 66 to the lane 79

Shairaz Baksh: This has been corrected.

  1. Lanes 133-145-318-323 change in symbol: NFB in NFkB

Shairaz Baksh: This has been corrected.

  1. Lane 179 control and change the ref “376” in superscript

Shairaz Baksh: We thank the reviewer for noticing this. This has now been corrected.

  1. Lane 221 add acronym of SB

Shairaz Baksh: This has been corrected. SB was not relevant here and has been removed.

  1. Lane 232 add reference of (Volodko et al, Cancers)

Shairaz Baksh: We thank the reviewer for noticing this. This has now been corrected.

  1. Lane 239 add acronym of CpGs

Shairaz Baksh: This has been corrected.

  1. Lane 259 add superscript of reference 26

Shairaz Baksh: This has been corrected.

  1. Lane 320 change TLR in TRLs

Shairaz Baksh: This has been corrected.

  1. Lane 360 change in symbol AMPK1

Shairaz Baksh: This has been corrected.

  1. Lane 384 change in symbol AMPK1

Shairaz Baksh: This has been corrected.

  1. Lane 572 change anti-TNF

Shairaz Baksh: This has been corrected.

  1. Lanes 575-577 some of the text is in bold, is it for some reasons?

Shairaz Baksh: This has been corrected.

  1. Lane 594 change in symbol AMPK1

Shairaz Baksh: This has been corrected.

  1. Literature

I checked the literature and noticed that the various works are a bit dated. There are only 3 quotes from 2021 and two from 2020, the others are older. Please update the literature.

Shairaz Baksh: We thank the reviewer for noticing this. This has been corrected and newer references have been added to some sections.

Reviewer 2 Report

Dear Authors Ia have read the manuscript and I send you my comments

1) Please delete line 147-151 these are results

2) Statistical analysis please add the power calculation

3) Please add the clinical data of the enrolled patients and the type of the disease

4) please add the inflammatory biomarkers in both blood and tissue: C reactive protein, white cell count, MMPs, and then you need to correlate these with clinical and histological data

Author Response

Reviewer Two

Dear Authors I have read the manuscript and I send you my comments

1) Please delete line 147-151 these are results

Shairaz Baksh: I looked at lines 147-151 and these lines are in the introduction and do not mention any results. Can the reviewer comment on this further if the need is still to remove?

2) Statistical analysis please add the power calculation

Shairaz Baksh: I looked at lines 147-151 and these lines are in the introduction and do not mention any results. Can the reviewer comment on this further if the need is still to remove?

3) Please add the clinical data of the enrolled patients and the type of the disease

Shairaz Baksh: We thank the reviewer for noticing this. We have now added this as table 1 “Demographics of patients in this study” and have clearly indicated how samples were utilized.

4) please add the inflammatory biomarkers in both blood and tissue: C reactive protein, white cell count, MMPs, and then you need to correlate these with clinical and histological data

Shairaz Baksh: We thank the reviewer for this comment. Unfortunately, not all patients had some or all of these markers stated in their charts. As such, we had access to only incomplete records and did not present these. Future studies will only access complete records and will be properly presented at that time. I hope this will be acceptable to the reviewer.

Round 2

Reviewer 2 Report

Dear Authors,

I send you again my comments: 

1) line 156: "We have collected over 500 blood samples and approximately  200 300 biopsy samples from pediatric, and adult IBD and non-IBD/control patients attending regular clinic appointments at the University of Alberta". 

This is a result not an experimental protocol, please change it

2) Power calculation is missing please add it

3) Table 1: please add clinical characteristic of the patients as previously required

4) Please add the requested data

Author Response

Response to Reviewers, Revision 2

Title: Novel Biomarkers for Inflammatory Bowel Disease and Colorectal Cancer: An interplay Between Metabolic Dysregulation and Excessive Inflammation

We thank the reviewers for their insightful comments and have made corrections to all comments. We hope these comments adequately answer your concerns.

Reviewer ONE

Reviewer one did not have comments after revision 1.

Reviewer Two

We thank the reviewer for reviewing our revision 1. We have added comments below that I hope will adequately answer the questions.

I send you again my comments: 

1) line 156: "We have collected over 500 blood samples and approximately > 300 biopsy samples from pediatric, and adult IBD and non-IBD/control patients attending regular clinic appointments at the University of Alberta". 

This is a result not an experimental protocol, please change it

Shairaz Baksh: We thank this reviewer for this comment. This has now been moved to section 3.1 (first results section).

2) Power calculation is missing please add it

Shairaz Baksh: We thank the reviewer for noticing this. We have now added power calculations to our statistical analysis: “The number of samples was determined using the sample size calculation: n =Z2 [p(1-p)]/(D2) and within the 90-95% confidence interval. Z equals the z-score [1.65 for 90% confidence interval and 1.96 for 95% confidence interval), p is the standard deviation and D is the margin of error. Using this power calculation, n = 106 -150 or greater within the 90-95% confidence interval.43” Although the reference is a dated 1965 it is well used even today. This formula and reference as established at the beginning of this study with consultation from patient oriented epidemiology experts at the Women and Children’s Institute at the University of Alberta.”

3) Table 1: please add clinical characteristic of the patients as previously required. Previous comment: Please add the clinical data of the enrolled patients and the type of the disease.

Shairaz Baksh: We have documented in Table 1 the disease type, sex and age of all patients recruited. We have added “Inclusion criteria for IBD patients was CD, UC, UC-CRC or CRC. Non-IBD excluded IBS, celiac or GERD patients. Exclusion criteria were patients with no co-morbidities unless for UC-CRC.” Lastly, we have added “Table 2: Disease sub-types in collected biopsies. Summary of endoscopic analysis at the time biospies were collected upon 121 chart reviews. “. I hope this is now acceptable to the reviewer.

4) Please add the requested data. Previous comment: please add the inflammatory biomrkers in both blood and tissue: C reactive protein, white cell count, MMPs, and then you need to correlate these with clinical and histological data

Shairaz Baksh: We thank the reviewer for this comment. Unfortunately, as mentioned previously, not all patients had some or all of these markers stated in their charts. As such, we had access to only incomplete records and did not present these. Future studies will only access complete records and will be properly presented at that time.

We have presented only the CRP and feacal calprotectin levels in supplemental Figure S2B for tissue levels of  RIPK2 vs blood levels of the biomarkers (text added to lines 457-458 “In addition, there appears to weak or no correlation between active RIPK2 and changes in c-reactive protein  (CRP) or faecal calprotectin (Supplemental Figure 2B)”).

The following legend was added to Supplemental Figure S2; “. B. Correlations between biomarkers. For left graph, r2 = 0.065 and p value 0.191, n = 31; right panel, r2 = 0.09 and p value – 0.278, n = 16.”  We did not perform analysis of CRP in tissues nor enough RIPK2 analysis I serum samples to obtain a robust powered correlation. I hope this will be acceptable to the reviewer.

Round 3

Reviewer 2 Report

Dear authors

thank you I have not comments